# Elementary Classroom Views of Nature Are Associated with Lower Child Externalizing Behavior Problems

**DOI:** 10.3390/ijerph20095653

**Published:** 2023-04-26

**Authors:** Amber L. Pearson, Catherine D. Brown, Aaron Reuben, Natalie Nicholls, Karin A. Pfeiffer, Kimberly A. Clevenger

**Affiliations:** 1Department of Geography, Michigan State University, East Lansing, MI 48824, USA; 2Department of Psychology and Neuroscience, Duke University, Durham, NC 27708, USA; 3MRC/CSO Social and Public Health Sciences, University of Glasgow, Glasgow G12 8QQ, UK; 4Department of Kinesiology, Michigan State University, East Lansing, MI 48824, USA; 5Department of Kinesiology and Health Science, Utah State University, Logan, UT 84322, USA

**Keywords:** visual green space, nature, trees, mental health, well-being, school, Michigan

## Abstract

Exposure to nature views has been associated with diverse mental health and cognitive capacity benefits. Yet, much of this evidence was derived in adult samples and typically only involves residential views of nature. Findings from studies with children suggest that when more greenness is available at home or school, children have higher academic performance and have expedited attention restoration, although most studies utilize coarse or subjective assessments of exposure to nature and largely neglect investigation among young children. Here, we investigated associations between objectively measured visible nature at school and children’s behavior problems (attention and externalizing behaviors using the Brief Problem Monitor Parent Form) in a sample of 86 children aged seven to nine years old from 15 classrooms across three schools. Images of classroom windows were used to quantify overall nature views and views of specific nature types (sky, grass, tree, shrub). We fitted separate Tobit regression models to test associations between classroom nature views and attention and externalizing behaviors, accounting for age, sex, race/ethnicity, residential deprivation score, and residential nature views (using Google Street View imagery). We found that higher levels of visible nature from classroom windows were associated with lower externalizing behavior problem scores, after confounder adjustment. This relationship was consistent for visible trees, but not other nature types. No significant associations were detected for attention problems. This initial study suggests that classroom-based exposure to visible nature, particularly trees, could benefit children’s mental health, with implications for landscape and school design.

## 1. Introduction

Globally there is a growing appreciation of the mental and physical health benefits of exposure to nature, including parks and forests (i.e., green space). Specifically, exposure to green space has been associated with lower depression, anxiety, aggression, stress and cortisol levels, a lower body and mass index, and higher physical activity and self-reported health status [1,2,3,4,5,6,7,8,9,10,11,12]. While most of this research has focused on the usage of, or access to, natural spaces [13,14], there is increasing evidence that visual (e.g., visible greenery) and even auditory exposure (e.g., birdsong) to nature may induce benefits, expressly for mental health [15,16,17,18,19,20].

Because of these potential benefits and the fact that 10% of children worldwide combat a mental disorder [21], numerous studies have analyzed how exposure to green space may protect children’s mental health, in particular [22,23,24,25,26,27,28,29]. Research on children’s green space exposure in their residential neighborhood has shown that a higher percentage of available green space leads to fewer behavioral problems [26,27]. Likewise, research from Barcelona detected protective effects of residential surrounding greenness on a range of socioemotional and behavioral symptoms including hyperactivity/inattention [2]. In addition, from the ages 7 to 12 y, children exhibited fewer conduct problems, decreased anxiety, and lower depression and somatization scores when living in homes with more surrounding greenery [27]. In a compelling longitudinal population study of the mental health benefits of green space, early life exposure to greenness (assessed via satellite imagery, thus indicative of higher levels of tree canopy) was protective against multiple disorders [23]. The risk for those living in areas with the lowest level of greenspace was up to 55% higher compared to those living in areas with the highest level of greenspace, even after adjusting for urbanization, socioeconomic factors, parental history of mental illness, and parental age. Importantly, this research also indicated that cumulative residential green space exposure during childhood showed a stronger association, compared to cross-sectional exposure. While most studies, including those outlined here, have shown mental health benefits of exposure to nature for children, not all findings are consistent. For example, other research did not find associations between behavior problems (e.g., hyperactivity/inattention) and residential surrounding greenness, although there was an association with residential distance to nearest urban green space in males specifically [23]. Another study unexpectedly found higher residential greenness was associated with more conduct problems and less prosocial behavior for children whose mothers had higher educational attainment [29]. While there are some studies with null or unexpected findings, the vast majority of studies underscore the potential for exposure to nature to benefit children’s mental health.

The benefits of exposure to nature appear to be differential by sex and socioeconomic status, although a recent review concluded that most such studies reported associations suggesting greenspace benefits on child behavior [30]. Longitudinal data from the Millennium Cohort Study showed that access to garden and use of parks and playgrounds were related to fewer conduct, peer and hyperactivity problems. Importantly, it appears that neighborhood green space may promote emotional well-being in poor urban children in early childhood, as those with more greenery had fewer emotional problems from age 3 to 5 y than their counterparts in less green neighborhoods [28]. Likewise, complementary research has shown that the relationship between greenspace and mental health in children differs depending on age. At younger ages (7 y), higher exposure to greenspace was significantly associated with lower externalizing behaviors, whereas exposure among older children (12 y) was associated with lower internalizing behaviors [27]. Thus, demographic characteristics may play an important role in the relationships between green space and mental health in children.

One pathway through which exposure to green space and nature influence children’s mental health may be through viewing natural scenes, or visual exposure. Naturalness and perceived restorativeness of visual scenes may be important for health because these emotions are thought to be linked to subconscious interpretation of scenes as either threatening or supportive. Specifically, natural features, through evolution, are associated with key resources, according to the Stress Recovery Theory [31]. Another theory, the Attention Restoration Theory, also supports the idea that natural scenes may be beneficial to mental health. This theory posits that natural features, in contrast to human-made features, do not require direct attention, thereby providing a calming backdrop for relaxation and restoration [32]. Indeed, laboratory evidence has shown an increase in parasympathetic nervous activity and lowered heart rate when viewing images of trees [33,34], a preference for more ‘natural’ images [35], and a preference for green and blue hues (e.g., sky) and ‘natural’ compared to built scenes [32,36,37]. In fact, in a study exploring people’s perceptions of apartment window views, views with maximum amounts of sky are perceived as most restorative [38]. Preference for sky views has also been identified as the primary factor in other studies [39,40]. Further, it is now understood that intrinsically photoreceptive retinal ganglion cells in the retina are sensitive to wavelengths of the color of the blue sky and exposure to these wavelengths provides essential signals to our circadian biological clock [41,42]. Disruptions in the circadian rhythm have cascading effects on mental health, including mood disorders and behavioral responses to stress [43].

While residential exposure to visible nature appears to be important, because school attendance is often compulsory and a large proportion of daytime hours are spent at school, school environments are a critical dimension of exposure to nature for children. In the U.S., most children spend approximately six hours a day for 180 days of the year in school settings (>1000 h each year) [44]. School-focused research has shown that students (ages 6–18 y) perform better academically when schools have higher levels of greenness within 250 m buffers around schools [45,46]. Other studies found that higher levels of green space (specifically trees) across campus were associated with improved attention recovery [47] and elevated scores on standardized tests [48,49], even after adjustment for wealth indicators (e.g., free lunch enrollment and socioeconomic status). Likewise, longitudinal research with a 12-month follow-up has shown enhanced working memory and reduced inattentiveness associated with greenness within and surrounding school boundaries [50]. It is important to note that greener school environments are not equitably distributed, and are often found in wealthier neighborhoods [51,52]. Most of the studies showing that school greenness is associated with better child outcomes have evaluated the entire school grounds, using aerial imagery normalized difference vegetation index (NDVI).

However, it is possible that children are primarily exposed to these large areas when outdoors for recess for short periods of the school day. Building on foundational research showing that hospital room views influenced recovery outcomes for surgery patients and the subsequent development of the Stress Recovery Theory [53], a glaring gap in our knowledge is how nature views from classroom windows (where children spend most of their school day) may influence mental health and behavioral outcomes. Of the few studies evaluating views of nature from classrooms, the quantification of nature views has been coarse and/or subjective (‘barely’ to ‘predominantly’) [54]. For example, studies have found that high school students exhibited increased educational attainment, better test scores and shorter stress recovery time when in classrooms with natural views compared to students with built-up views or no windows [49,55,56]. Similarly, students (12 to 20 y) with green space present at school reported higher perceived restorativeness, after adjusting for age, sex, accommodation, and income [54]. While these studies are important in highlighting the potential role of classroom nature views on children’s mental health and behaviors, the quantification of nature views are not readily reproducible or at a fine resolution. Such studies have also not yet been conducted among younger children or on many health outcomes and behaviors. 

The current study aimed to overcome several research gaps by: (i) objectively quantifying nature views at the classroom window scale, (ii) utilizing a valid measure of child behavior problems, (iii) accounting for residential views of nature, and (iv) including a sample of young children. We specifically utilized attention and externalizing behaviors to evaluate associations with nature views because of the role these problem types have in both diagnosis of childhood mental illnesses as well as long-term negative consequences. Specifically, attention and externalizing behavior problems have been shown to lead to increased number of referrals to mental health centers for children with cascading effects on deleterious adult outcomes (i.e., occupational rank, job performance, and social skills) [57,58,59,60]. Thus, understanding factors that influence children’s attention and externalizing behavior problems may yield far-reaching benefits. 

In this study, we hypothesized that higher exposure to visible nature from the classroom window will be associated with lower attention and externalizing behavior problem scores, after accounting for nature views from home and other demographic characteristics. We further hypothesized that tree and sky views from classroom windows would preferentially associate with lower problem behavior scores (demonstrating larger association sizes than views of grasses and shrubs). 

## 2. Methodology 

### 2.1. Sample 

Children from 15 first- (*n* = 7) and second-grade (*n* = 8) classrooms at three elementary schools in East Lansing, Michigan, were assessed in May 2019 (late spring/early summer). The school term runs September to June; assessments occurred near the end of the students’ school year. In the early years of elementary education, most schools do not have children move between classrooms for different subjects (with exceptions for music and physical education), so children spend the majority of their learning time in their assigned classroom. We first selected all elementary schools with planned, future renovations to their schoolyard (*n* = 3). Then, all first- and second-grade classrooms were invited to participate in this study via contact with the principal and then each teacher. Students were then recruited via materials about this study and consent forms to guardians. After consent, parents and guardians reported on child demographic details (age, sex, and race/ethnicity), and child behavior (described below) via printed survey materials distributed and returned to classrooms. School socioeconomic status/quality was measured using the Michigan Center for Educational Performance and Information [61] “Michigan School Index”, which ranks schools on eight measures of student proficiency and accomplishment, including attendance and graduation rates, advanced coursework completion, postsecondary enrollment, and staffing ratios. Schools are ranked on a scale of 0, low performing, to 100, high performing. The three schools in this study are public institutions located within residential neighborhoods. School 1 had the largest number of students at the time of this study (*n* = 368), followed by School 2 (*n* = 346), and School 3 (*n* = 316). Human subjects’ approval was granted by Michigan State University (STUDY00002552). Parents/guardians provided written consent and children provided written assent (*n* = 90 recruited). The sampling frame and study design follows past studies of children in schools involving primary data collection on outcomes, e.g., [50] (i.e., school selection then population sample of students). 

### 2.2. Quantification of Nature Views at School

To quantify views of nature from classroom windows, we first captured digital images of each window within each of the classrooms upon one visit in June 2019. We did not alter any of the classroom conditions, including the positions of window blinds, décor, and furniture. All photos were taken during the same season, within two days of each other. The images were then uploaded into Photoshop (Adobe Photoshop CS. 2004, Berkeley, CA, USA), where each was cropped, and the image size was uniformly set to 1200 × 1500 pixels. Following this, the images were transferred to ArcMap v10.6 (ESRI, Redlands, CA, USA). Within each image, polygons were constructed to encompass specific types of nature (grass, tree, sky, shrub) (Figure 1). Total pixel counts for these nature types were calculated. Then, percentages were tabulated for each nature type before being combined to determine the total nature visible from each window. Percentages were then averaged for each classroom. 

Because we were concerned with obstruction such as blinds influencing the quantification of nature, in sensitivity analyses, we determined the percentage of the window that was unobstructed by blinds, where a higher percentage indicates less of the view was obstructed.

We also calculated the nature visible across the school grounds (to compare with most of the existing studies) using aerial imagery of Normalized Difference Vegetation Index (NDVI). Imagery were compiled for each school using Landsat 8 data (resolution 30 m, June 2019). A 250-m buffer around the centroid of each school (DTMB- Michigan Schools and Districts, or when unavailable via Google Earth) was constructed around each school. This buffer size was used to parallel other research on school ground greenness [45,46]. The average NDVI value for all cells within each buffer was calculated for each school. Last, we also were concerned about potential differences in nature exposure based on the number of windows per classroom. So, we also created a measure of nature exposure by summing each window’s percentage nature view. 

### 2.3. Quantification of Nature Views at Home and Residential Deprivation Score

To account for nature exposure at home, we utilized Google Street View (GSV) imagery to obtain images for every child’s residential view from the front door (i.e., 180° facing away from the front door). Each image was first standardized to 1500 × 1200 pixels and imported into ArcMap where polygon creation was conducted and percentages by type of nature and overall were calculated in an identical process as the classroom view images (Figure 2). We also linked home location with the Area Deprivation Index at the blockgroup-level, constructed using American Community Survey estimates from 2016 to 2020 [62], where scores indicate high (10) to low (1) deprivation relative to all blockgroups in Michigan.

### 2.4. Assessment of Child Behavior

Children’s behaviors were reported on by a guardian (*n* = 81) or teacher (*n* = 9) using 18 items on the Brief Problem Monitor–Parent Form (BPM-P, ASEBA, University of Vermont), which indexes behavior problems in the domains of internalizing, externalizing, and attention problems. (Teachers were utilized in cases where guardians could not be reached. Removing teacher-rated children did not alter the results.) The BPM-P asks guardians to rate children’s behavior over the past 30 days (0 ‘not true’, 1 ‘somewhat true’, 2 ‘very true’) on items assessing behavior problems, such as ‘arguing a lot,’ or ‘is disobedient at school’. Items are summed within each domain to produce separate internalizing, externalizing, and attention problem scales, as well as an overall “total problems” scale. These scales have been widely validated in previous research and demonstrated good internal consistency (Cronbach’s α = 0.86–0.87) [63]. Because our sample includes young children (ages 7 to 9 y), we specifically hypothesized that externalizing behaviors would be influenced by nature views [27]. Because of the importance of attention for learning at school and building on prior research showing associations between greenspace and attention problems [30,47,50], we also hypothesized that nature views at school would be associated with lower attention problems. Thus, our primary analyses were restricted to these outcomes. In our sample, Cronbach’s α was 0.89 for attention problems (6 items) and 0.80 for externalizing problems (7 items). All subsequent analyses were conducted on children for whom we had complete data for address, demographics, and behaviors (*n* = 86).

### 2.5. Statistical Analyses

Statistical analysis proceeded in three stages. First, descriptive statistics were calculated on key study variables. Second, to evaluate the primary hypothesis, we fitted two Tobit regression models (one for each behavior outcome) regressing the problem behavior scales onto classroom nature view. Each model included total nature viewed from the classroom as the independent variable and a behavior problem score as the dependent variable, adjusting for child age, sex, race/ethnicity, and nature exposure level at home. Tobit regression was used to account for the bounded or censored nature of the outcomes measured as a scale, as done in other research involving scales [64]. We evaluated the need to run multilevel regression and found that all intraclass correlations (ICCs) for school and class were very small (<0.0001). Thus, multilevel modeling was not warranted. 

Third, to evaluate the secondary hypothesis, that tree and sky views from classroom windows would show stronger associations with problem behavior scores than views of grasses and shrubs, we fitted similar regression models for each component type of nature viewed from the classroom (grass, tree, sky, and shrub) and each behavior problem score separately, adjusting for the same covariates. Given the large number of statistical tests conducted on the secondary hypothesis, secondary results should be viewed as exploratory and interpreted with caution. 

We also conducted several sensitivity analyses. First, we re-ran all models while accounting for percentage of windows that was unobstructed by blinds in the model. We observed a slight attenuation of the externalizing behaviors–visible trees association and no changes in the total nature views association (Appendix A). We also observed that lower levels of obstructions (higher values) were significantly associated with higher externalizing behavior problem scores. Second, we repeated all study tests substituting school-ground average NDVI in place of classroom nature views as the independent variable of interest in the tests of child behavior problems (Appendix A). No school-ground tests were statistically significant. Third, we also were concerned about potential differences in nature exposure based on the number of windows per classroom. So, we also created a measure of nature exposure by summing each window’s percentage nature view. This measure did not yield significant associations with behavior scores (Appendix A). Fourth, for transparency and comparison to other research, we also fitted regression models for internalizing behaviors and total behavior problems, but did not detect significant associations (exploratory results found in Appendix A). Fifth, as a final sensitivity analysis, we fitted two regression models (one for each behavior problem score) where residential nature exposure was the independent variable of interest, and adjusted for age, sex, and race/ethnicity compare with findings related to classroom views (Appendix A). No significant associations were detected in this sensitivity analysis. All statistical analyses were conducted in Stata v16 (College Station, TX, USA).

## 3. Results 

Study children were primarily female (75.6%), White (80.2%), and, on average, eight years old (range age 7–9) (Table 1). On average, children demonstrated modest, non-clinical levels of externalizing and attention problems (mean = 2.4 and 2.9 on a scale of 0 to 14 and 0 to 12, respectively), levels consistent with normative samples of children lacking psychological diagnoses [65]. At home, children tended to have high levels of nature exposure (71% nature view from home windows on average) and lived in neighborhoods with comparable scores in neighborhood deprivation (mean = 4.7 on a scale of 1 to 10).

Study children’s classrooms had between 2 and 7 windows in total (overall mean 4 windows, SD = 1) with school 3 having lower numbers of classroom windows on average (2, SD = 0) than school 1 and 2 (mean = 5 and 6 windows, respectively) (Table 2). Children’s views from these windows were primarily of non-natural objects or surfaces (mean = 69%). On average, 31% were ‘nature views’, which included trees (mean = 14.3%, range 17–55), shrubs (mean = 5.6%, range 0–23), sky (mean = 4.7%, range = 0–17), and grass (mean = 3.6%, range = 0–16). The vegetation included deciduous and evergreen trees and shrubs, and turf grass (see Appendix A for images).

On average, the percentage of natural view was comparable at school 1 (24.6%) and school 2 (24.9%), while school 3 was greater (44.6%) (Table 2). The average level of vegetation (NDVI) across all three school grounds was moderately high [66] (means > 0.35). All schools were on the higher end of school quality/performance (School Index Scores from 87.9 to 97.8) [61]; no school met state cutoffs for ‘underperforming’. Child behavior problem scores differed across schools, with school 2 demonstrating the highest attention and externalizing problem scores and school 3 demonstrating the lowest scores. 

### Nature Exposure and Behavior Problems

Children attending classrooms with greater nature views demonstrated fewer behavior problems than their peers attending less nature-rich classrooms (Figure 3). After adjustment for age, sex, race/ethnicity, residential deprivation score, and nature exposure at home, for every 10% higher classroom nature view, children demonstrated a 1.2 unit lower score in externalizing behaviors (standardized β = 0.58, *p* = 0.009). After adjustment for study covariates, total nature view was not, however, associated with lower attention problems (standardized β = 0.05, *p* = 0.780).

Tests of the secondary hypothesis, that trees and sky views would show larger associations with behavior problem scores than grass and shrub views, indicated that views of trees likely accounted for the statistical association of total nature view with behavior problems. Specifically, after adjustment for study covariates, for every 10% increase in tree views, children demonstrated a 0.08 unit (standardized β = −1.02, *p* = 0.009) lower score in externalizing behaviors. There were no statistically significant associations of the other nature view types (grass, shrub, sky) with externalizing behaviors. Consistent with our primary study findings on total nature view, no nature subtypes were significantly associated with attention problems. 

## 4. Discussion

Exposure to nature, including visibility of nature, may be an important source of support for mental health [20,67,68]. Visual exposure to nature through classroom windows may be particularly relevant for children in school settings because children spend extended periods of time at school, mostly indoors, and may have limited opportunities to physically engage with nature outdoors. Our study’s research objectives were based on this. First, we sought to test the relationship between visible nature from classroom views and children’s behavior problems. Additionally, we wanted to separately evaluate specific types of nature and behavior problems. As a sensitivity analysis and to comprehensively consider other aspects of nature views, we also quantitatively assessed nature views across the school grounds and at home. In doing so, we found that classroom nature views, and trees specifically, were negatively associated with children’s externalizing behaviors (but not attention problems). In contrast, there were no significant associations with residential views or across school grounds. 

We found that children spending their days in classrooms with greater average (per window) views of nature were rated by their guardians as demonstrating fewer externalizing behavior problems, at home and school. This finding aligns with several studies showing associations with complementary outcomes including learning attainment, mental recovery, and graduation rates [46,49,55,56]. It appears that these results may be driven, in part, by the presence of trees, as found in other research [33,34]. Indeed, other research that found specific types of nature on school campuses, in particular trees and shrubs, were beneficial to students [48,49]. Because trees represent vertical forms of green space, they may be more easily viewed when either nearby or more distant from windows than horizontal forms (e.g., grass). Since we used the percent of the classroom view covered in trees, our findings suggest that either more distant trees or fewer but nearby trees may be salutary for children’s behavior. In addition, trees provide a distinctive environment which can increase natural sounds, especially birdsong [68,69], which has been shown to influence positive affect and decrease stress and annoyance [16], as well as aid in temperature control and the provision of shade. 

We did not, however, find significant associations with behavior problems and sky views from classrooms. This could be, in part, related to generally low levels of sky views captured through windows and this form of imagery. Increasing the number of windows assessed and the heterogeneity of views may allow more robust evaluation of our finding. This finding is similar to a Swedish study that found that young children at schools with high numbers of large trees, but not sky views, had lower levels of attention problems [70]. However, it is in contrast to studies reporting a preference for sky views from windows [39,40] and for the importance of sky views for regulating the circadian rhythm [41,42]. However, if children spend adequate amounts of time outdoors, they may be exposed to ample daylight for biological regulation, regardless of classroom views.

Beyond the classroom window views, we did not detect significant, independent associations between the school grounds NDVI with our outcomes. While no other studies to date have evaluated the effects of school grounds greenness explicitly on externalizing behaviors, our findings differ from other studies that found that total greenness of the school grounds were associated with academic attainment and performance [45,46]. Our null findings related to school grounds NDVI may be an artefact of the small number of schools included in this study and the lack of variation in NDVI across these three schools. Thus, further investigation is warranted that includes a larger number of schools and diversity in the school grounds settings. 

Likewise, our findings are in contrast to other research that showed the importance of residential green space on mental well-being, and cognitive development of young children [71], particularly for girls [72]. In fact, of the few studies examining associations between green space and externalizing behaviors in children, all evaluated residential green space. One study of young children (7 y) found protective effects of residential greenery on conduct problems, but not other internalizing problems [27]. In contrast, in older children (10 y), residential green space was associated with lower hyperactivity/inattention in boys [28] (but not conduct problems). Among adolescents (9–18 y), both short-term (<6 months) and long-term (1–3 years) exposures to residential greenspace were associated with reduced aggressive behaviors. However, most studies assessed nearby nature using measures such as distance to nearest park or NDVI within a buffer of the home, rather than greenness at eyesight level from the front of the home. There are a few notable methodological differences which may explain these disparate findings and warrant further investigation to clarify the relationship between residential nature and child-level outcomes. Namely, the novel approach used in the present study focused on overall nature viewed from the child’s front door, not greenness and/or the surroundings. 

Another potential reason for inconsistent findings could be that child externalizing outcomes may be affected by green space but that the importance of exposure setting (school versus home) may change as the child develops. Extant research has shown that neighborhood conditions influence externalizing behaviors and these links appear early in life and importantly, increase over time [73]. This may relate to increases in agency starting in middle childhood and adolescence [74], whereby adolescence become more mobile and autonomous, yielding more direct exposure to their neighborhood environments. In contrast, younger children may experience fewer neighborhood exposures in comparison to the school environment. This is underscored by our significant findings for classroom views on externalizing behaviors for young children, in particular.

### Strengths and Limitations

By quantifying visible nature from classroom windows, we were able to evaluate associations with behavioral outcomes using methods that could be easily replicated in other research to compare with our findings. While the photos were uniform in size (i.e., pixel count), the actual size of each window in each classroom was not evaluated. We also considered the overall greenness of the school grounds using NDVI data, but this measure is at a coarser scale than the methods used to assess nature views from classroom windows and at home in this study. To utilize more comparable methods, future studies could capture eye-level views using imagery at multiple locations. For example, images could be collected from the cafeteria, library, or gymnasium windows or 360° images could be taken on the playground. In doing so, views of nature across the school grounds may be more realistic for the experience of a child. These data could also be linked to GPS, eye trackers, and/or wearable cameras worn by children to quantify cumulative nature views exposures [75]. In this study, we did not account for amount of time each child spent in their classroom, working under the assumption that this would not vary greatly from classroom to classroom or between schools for first and second graders. However, future research to calculate cumulative exposure could also include time spent in view of any natural environment, including those at school, home, or elsewhere (e.g., neighborhood parks). In addition, the NDVI 250 m buffer size presents a limitation in that the school buildings with larger area would inherently have a lower NDVI values in the buffer, as building surfaces yield lower NDVI values. However, in our study, the entirety of each school ground, as well as some of the surrounding area, fit within the 250 m buffer size. 

Other limitations include that we did not measure daylight from the windows, which may be a more accurate measure related to circadian rhythm regulation, nor did we account for which direction a particular child sat in the classroom, which window might be visible, or whether children were visually impaired. This means that we may be over- or under-estimating the nature view exposure for a given child. Similarly, we do not know the location of the child’s bedroom at home or where they spend much of their time. Therefore, an evaluation of nature available from the front of the home may under- or over-estimate children’s residential nature view exposures. This may be particularly problematic for apartment buildings, although few participants resided in this setting in the present study (*n* = 14). This may be an important consideration for future studies where the population typically resides in apartment or multifamily dwellings. 

Last, this study was observational and cross-sectional, and thus causal inference is inappropriate. Future studies could follow an experimental design, where children are assessed before and after an intervention (e.g., increasing the visible nature from the classroom, transferring schools, and/or moving homes). Alternatively, researchers might choose to study the same set of children across time to determine how various classroom views affect children’s behavior problems and if there are particularly critical times in child development or specific subpopulations who may benefit the most. Our study was conducted in early summer, when deciduous trees have leaves and many plants are at the height of their growth and greenness. Other studies may wish to examine seasonal differences or specific types of vegetation that yield benefits. Further, the relatively small sample size used in this study may have influenced our ability to detect significant associations as we were likely underpowered to detect very small effects. The relatively large effect size reported for the main study finding may also reflect imprecision in the effect parameter driven by the low sample size [76]. We would expect the point estimate for the association of nature view with externalizing behaviors to decline with larger samples in future studies. Furthermore, this study was conducted using data from Michigan, USA, located in a continental climate zone. Studying different regions of the globe could increase the validity of our findings. Though we cannot claim causation, this study provides a strong foundation for future projects to proceed by showcasing a new methodology to quantify views of nature from classroom windows and the potential importance of school green space on externalizing behaviors in young children. 

## 5. Conclusions

The purpose of this study was to evaluate associations between visual exposure to nature at school and children’s behavior problems. We found that nature views from classroom windows, and trees in particular (although the latter was exploratory analysis and should be interpreted with caution), may be important sources of support for lowering externalizing behaviors among young children. Children spend many of their waking hours at school, and as such, access to visible nature from classroom windows may be particularly beneficial. This research employed the use of an innovative method of objectively quantifying nature views which has clear implications for school design. When schools are considering new or renovations of school grounds, the integration of additional trees in locations that can be viewed from classrooms may serve as a beneficial resource to children. Future research could usefully employ an experimental design, include a larger and more heterogeneous sample of schools and children, and include imagery from multiple locations where children spend their time at school.

## Figures and Tables

**Figure 1 ijerph-20-05653-f001:**
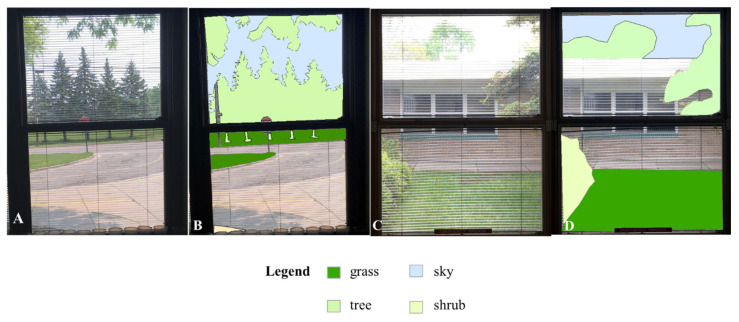
Examples of original images from classroom (**A**,**C**) and images with quantified nature as polygons (**B**,**D**).

**Figure 2 ijerph-20-05653-f002:**
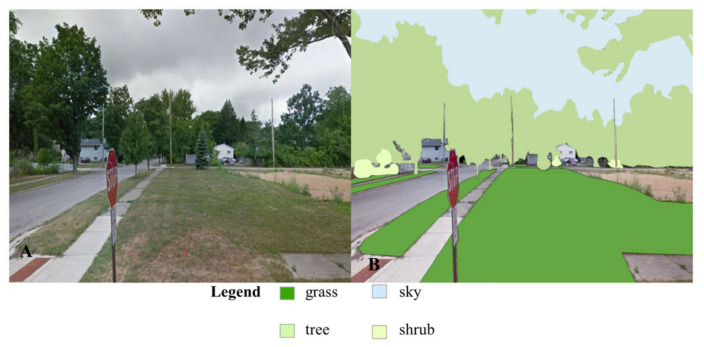
Example of an original Google Street View image (**A**) and an image with quantified nature as polygons (**B**).

**Figure 3 ijerph-20-05653-f003:**
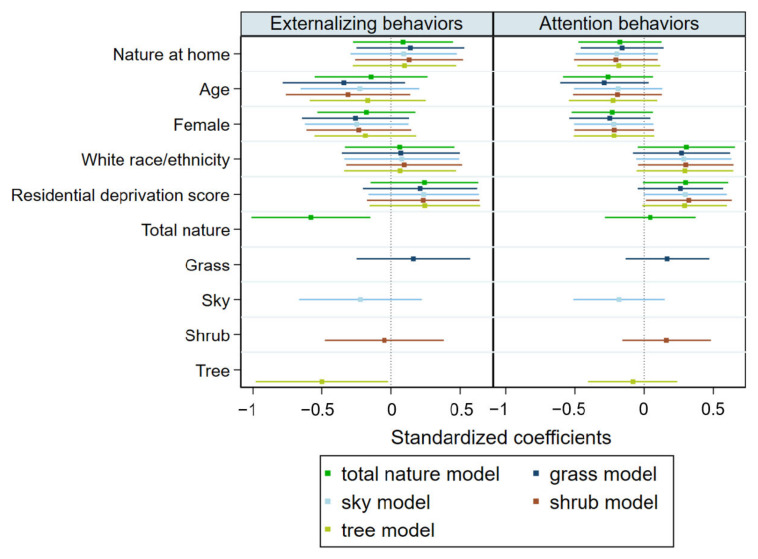
Separate regression modeling results predicting externalizing behaviors and attention problems with classroom views of nature.

**Table 1 ijerph-20-05653-t001:** Descriptive statistics for the sample of children by school and overall.

	School 1	School 2	School 3	Overall
	*n* = 53	*n* = 18	*n* = 15	*n* = 86
Age, mean (sd)	7.7 (0.7)	8.1 (0.7)	7.8 (0.8)	7.8 (0.7)
Race/ethnicity, % White	79.2	83.3	80.0	80.2
Female, %	67.9	88.9	86.7	75.6
Attention problems, median (mean)	1.0 (2.8)	3.0 (3.4)	1.0 (2.5)	2.0 (2.9)
Externalizing problems, median (mean)	2.0 (2.5)	3.5 (3.2)	0 (1.0)	1.5 (2.4)
Residential nature view, mean (sd)	70.3 (12.1)	72.5 (11.7)	69.9 (13.5)	70.7 (12.2)
Neighborhood deprivation score, mean (sd)	4.6 (1.8)	5.3 (2.2)	4.5 (2.7)	4.7 (2.1)

Attention Problems on scale 0–12; Externalizing Problems on scale 0–14; sd: standard deviation.

**Table 2 ijerph-20-05653-t002:** Descriptive statistics for school-based nature exposure by school and overall.

	School 1	School 2	School 3	Overall
Classrooms	*n* = 6	*n* = 5	*n* = 4	*n* = 15
School index score	87.9	96.9	97.8	94.2
Children per classroom, mean (sd)	25.7 (3.8)	23.6 (5.5)	23.8 (3.0)	24.5 (4.1)
Windows per classroom, mean (sd)	4.7 (2.1)	5.8 (0.4)	2.0 (0.0)	4.1 (0.9)
School grounds NDVI, mean	0.359	0.394	0.403	0.374
Classroom nature view %, mean (sd)	24.6 (6.7)	24.6 (6.7)	44.6 (6.7)	31.1 (11.0)
**Type of classroom nature view**				
Grass %, mean (standard error)	1.3 (0.2)	11.1 (1.3)	2.7 (0.6)	3.6 (0.5)
Tree %, mean (standard error)	10.2 (1.1)	8.9 (1.4)	35.4 (2.9)	14.3 (1.4)
Sky %, mean (standard error)	5.8 (0.8)	4.7 (0.7)	0.5 (0.1)	4.7 (0.6)
Shrub %, mean (standard error)	7.3 (1.4)	0.2 (0.1)	6.0 (2.6)	5.6 (1.0)

NDVI: Normalized difference vegetation index, conducted using 250 m buffer around school centroid; sd: standard deviation.

## Data Availability

Data are unavailable due to privacy or ethical restrictions.

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
