# Peer review of "Elementary Classroom Views of Nature Are Associated with Lower Child Externalizing Behavior Problems"

_ijerph, 2023, doi:10.3390/ijerph20095653_

Round 1
Reviewer 1 Report
Thank you for the opportunity to review your manuscript, which reports a correlational study exploring associations between children’s behaviour problems and school classroom window views while controlling for demographic factors and residential nature views. The finding of a negative association between school classroom window views and externalizing behaviour problems for the nature type of visible trees must be taken into consideration of other potential factors and the relatively small sample. Nonetheless, the manuscript introduces a novel method for objectively measuring visible nature at school, which makes it a useful contribution to the literature.
I have inserted highlights within the manuscript either to indicate where attention needs to be directed to wording, or to flag some specific comments or queries for you to take into consideration. These are detailed below.
Introduction
Lines 108-109. “While residential exposure to visible nature appears to be important, the compulsory nature and time children spend in school settings indicates that school environments are a critical dimension of exposure to nature.” The wording here is confusing because indicating ‘compulsory nature in school settings’ and ‘time spent in school settings’. Are you suggesting there is a ‘compulsory nature’ component of school settings? Or do you perhaps intend to refer to the fact that schooling itself is compulsory?
Line 113. There is referent to findings of better academic performance “when schools have higher levels of greenness”. We need to acknowledge the importance of controlling for socioeconomic factors. For instance, Baró et al. (2021) reported that "A substantial share of greener schools are located in the wealthiest neighborhoods" https://doi.org/10.1016/j.landurbplan.2020.104019 and van Velzen and Helbich (2023) reported that "Greener school outdoor environments tended to be in wealthier neighborhoods" https://doi.org/10.1016/j.landurbplan.2023.104687.
Line 133. You claim that “students (12 to 20y) with green space present at school reported better self-rated health and quality of life” with a citation of Akpinar in support of this claim. However, Akpinar reported that "none of the health indicators was correlated with high school greenness". It was only students' perceived restorativeness that was predicted by presence of green space in the school campus. The differences in health indicators were between younger and older students, not associated with greenness.
Materials and Method
Line 154. The data were collected in May 2019. At that time of the school year, how much time had students then spent in their respective classrooms? Was it just the specific classroom that each class attended at that school (i.e., didn't move to different rooms for different subjects)?
Lines 235-236. “All subsequent analyses were conducted on 235 children for whom we had complete data for address, demographics, and behaviors 236 (n=86).” It’s best to include a statement to confirm that the sample size is sufficient for the type/s of analysis performed on the data with respect to power. What would be an optimal sample size to detect effects?
Results
When reporting results, avoid repeating content from tables within text.
Discussion
Section 4.1 Strengths and limitations. The duration of time students had spent in classrooms up to the point of data collection should have been reported earlier, and depending on how long that might have been, it could also be a limitation which should be acknowledged here. I agree that time and exposure to nature at other locations at school should also be considered as an influencing factor, as should opportunities to engage in nearby nature experiences at locations other than school or home (e.g., neighbourhood parks).

Author Response
See attached response letter, please.

Reviewer 2 Report
Dear Authors
I enjoyed reading your work. Overall, it is well written and presented.
I want to raise a few points that I believe will inform readers about the "best building design/landscape for pupils" and also inform further research.
Please provide in the methodology a description of each school's view from the classrooms (including the type of planting - please see below), as well as the size of the windows in relation to the size of the wall. This information can potentially provide guidelines for building betters schools and schoolyards.
Furthermore, the research was undertaken in May. What kind of planting was seen from the windows? Were the plants / trees in bloom? What color was the tree foliage? What shape was the canopy? There is no detailed description of the planting or the trees of the schools. If the research was undertaken in fall (deciduous trees would be colorful) or in winter (deciduous trees would be bare) would the results be the same? Are evergreen trees (i.e. the same year-round green) better than deciduous trees (i.e. potentially more stimulating by seasonal change)? There is research on the effect of seasonal changes in planting on people's health/behavior. The qualitative traits of plants/trees, although not studied per se, should be addressed and mentioned as they provide information for further research that potentially can lead to reducing child externalizing behavior problems.
Kind Regards
Author Response
See attached response letter, please.

Reviewer 3 Report
Dear authors,
thank you for the opportunity to review your interesting work on nature and behavioral problems of elementary school children. I enjoyed reading this work.
The introduction is very detailed and leads the reader well to the aim and objectives of this paper. The research gaps and and merit and relevance of the study are very clearly presented well done. The only aspect that I may suggest, and keep note this may be matter of disciplinary differences, is if a seperate literature review could be presented. I understand it is at present part of the introduction, but it may ease the readers life if the factors related to attention and externalizing behaviors and nature views could be presented seperately.
Please consider to revisit the aim and objective statement and cut out redundant information that may belong in the method section.
Can the authors in the method section clarify how many children participated in the study. This is a bit unclear and gets confusing due to the number of schools mentioned.
Can the authors indicate if the study design follows prior studies and provide reasoning for this. The nature quantification section is suffienctly described and the reader can follow along well.
The authors indicate: Students were then recruited via materials about the 156 study and consent forms to guardians. So can you make more explicit is this a convinience or a purposive sample? Please justify the appropiateness.
In the analysis section- you mention a hypothesis. Can you please indicate the hypothesis along with the framework of the study
Can you please justify the appropiateness of the tobit model.
Can the results and discussion section be merged
Excellent conclusion section, clear and reflective work
Author Response
See attached response letter, please.
